# Mapping the Burden of Fungal Diseases in the United Arab Emirates

**DOI:** 10.3390/jof10050353

**Published:** 2024-05-15

**Authors:** Fatima Al Dhaheri, Jens Thomsen, Dean Everett, David W. Denning

**Affiliations:** 1Department of Pediatrics, College of Medicine and Health Sciences, United Arab Emirates University, Al Ain P.O. Box 15551, United Arab Emirates; 2Department of Public Health and Epidemiology, Khalifa University, Abu Dhabi P.O. Box 127788, United Arab Emirates; jtw.thomsen@web.de (J.T.); dean.everett@ku.ac.ae (D.E.); 3Infection Research Unit, Khalifa University, Abu Dhabi P.O. Box 127788, United Arab Emirates; 4Manchester Fungal Infection Group, The University of Manchester, Manchester Academic Health Science Centre, Grafton Street, Manchester M13 9NT, UK

**Keywords:** invasive candidiasis, vulvovaginal candidiasis, aspergillosis, asthma, AIDS, keratitis

## Abstract

The United Arab Emirates has very little data on the incidence or prevalence of fungal diseases. Using total and underlying disease risk populations and likely affected proportions, we have modelled the burden of fungal disease for the first time. The most prevalent serious fungal conditions are recurrent vulvovaginitis (~190,000 affected) and fungal asthma (~34,000 affected). Given the UAE’s low prevalence of HIV, we estimate an at-risk population of 204 with respect to serious fungal infections with cryptococcal meningitis estimated at 2 cases annually, 15 cases of *Pneumocystis* pneumonia (PCP) annually, and 20 cases of esophageal candidiasis in the HIV population. PCP incidence in non-HIV patients is estimated at 150 cases annually. Likewise, with the same low prevalence of tuberculosis in the country, we estimate a total chronic pulmonary aspergillosis prevalence of 1002 cases. The estimated annual incidence of invasive aspergillosis is 505 patients, based on local data on rates of malignancy, solid organ transplantation, and chronic obstructive pulmonary disease (5.9 per 100,000). Based on the 2022 annual report of the UAE’s national surveillance database, candidaemia annual incidence is 1090 (11.8/100,000), of which 49.2% occurs in intensive care. Fungal diseases affect ~228,695 (2.46%) of the population in the UAE.

## 1. Introduction

Emerging infectious diseases caused by fungal pathogens have been recognized as a serious threat to the ecosystems of our planet, with tremendous impacts leading to multiple species extinctions, food insecurity, and geo-ecological disturbances [1]. The impact on human life is also substantial, though often overlooked with over a billion people globally estimated to be affected by fungal diseases resulting in more than 2.5 million attributable deaths annually [2]. Termed the “Hidden killers” [3], fungal diseases exceed mortality rates of tuberculosis on a global scale and malaria by more than 3-fold [2]. The designation of “One Health”, first coined during the emergence of SARS coronavirus and H5N1 influenza in 2003–2004 [4], is a call for a concerted effort to include human, animal, and plant health as a unified, trans-disciplinary front in global public health measures. This call is rooted in the recognition that the evolution and impact of antimicrobial resistance (AMR), including antifungal resistance, is influenced by every component that makes up our ecosystems [5].

The main drivers of mortality from fungal diseases are invasive fungal infections (IFI) in immunocompromised patients and those in critical care, which often exceed 50% in these populations despite the availability of antifungal treatments [3]. Chronic pulmonary aspergillosis (CPA) and fungal asthma undoubtedly also contribute substantially to the global burden of fungal disease. In patients with Human Immunodeficiency Virus (HIV), many deaths that arise from acquired immunodeficiency syndrome (AIDS) are attributable to opportunistic fungal infections. Paradoxically, the immunocompromised patient population continues to grow with the advances that are made in novel chemotherapeutic agents and immunomodulators. Hundreds of millions of people have chronic respiratory diseases (primarily COPD and asthma) and although the absolute risk of invasive and life-threatening infection is relatively low, the numbers affected dwarf those with immunocompromised conditions [6,7,8]. IFIs may directly kill immunocompromised patients or delay life-saving measures for their underlying life-threatening conditions. Apart from their underlying immunodeficiency, delays in instituting appropriate antifungal treatment due to a lack of accessible, accurate, and rapid fungal diagnostics further contribute to the high mortality rates in these vulnerable populations [9,10,11].

IFI has been greatly implicated in deaths during the COVID-19 pandemic, with three groups of fungal pathogens identified to be worsening outcomes when causing co-infections in COVID-19 critically ill patients [12]. COVID-19-associated pulmonary aspergillosis (CAPA), frequently caused by the azole-resistant *Aspergillus* species, has been shown to independently contribute to more than 40% of estimated mortality rates amongst COVID-19 critically ill patients [12,13,14]. Reported in the lay press as the feared “Black fungus”, COVID-19-associated mucormycosis (CAM) was declared an epidemic by the government of India during the second wave of the pandemic, with over 100,000 cases nationwide—an estimated burden reported to be 70 times higher than the global rate with a harrowing mortality rate [15]. While candidaemia is a frequent complication in intensive care unit (ICU) patients, a two- to ten-fold higher incidence has been reported in patients with COVID-19, including *Candida auris*, which results in a 30–83% mortality rate [16,17].

With the drastic human loss associated with fungal diseases comes a deeply felt economic impact. The US economic burden of fungal infections was conservatively estimated to be USD 11.5 billion in 2019, with USD 7.5 billion of direct medical costs alone, USD 3.2 billion attributable to premature deaths, and USD 870 million in productivity loss due to absenteeism. These statistics are most likely underestimates given the challenges in diagnosis and correct reporting [18].

Despite these alarming indicators, fungal diseases and the field of clinical mycology remain underfunded and underappreciated, with many public health agencies worldwide having no programs for mycological surveillance [3]. The World Health Organization (WHO) published its first-ever report in October 2022 highlighting a list of fungal “priority pathogens”, which includes 19 fungi that represents “the greatest threats to public health” [19]. This is the WHO’s first effort to prioritize fungal infections, having had no formal program addressing this growing global threat.

The Leading International Fungal Education (LIFE) program has aided in the estimation of fungal disease burden for over 6.8 billion people through publicly accessible published literature and abstracts worldwide (Figure 1) [20]. These estimations revealed differences in the global burden between countries, within regions of the same country, and amongst the different at-risk patient populations. Despite this impressive feat, many regions in the Middle East, North Africa, and Turkey (MENAT) remain under-mapped or not mapped at all, including the United Arab Emirates (UAE), due to the lack of published data [2,21]. The currently available literature from the MENAT region that explores fungal burden mapping includes published studies conducted in Qatar, Oman, Kuwait, Saudi Arabia (abstract form only), Jordan, Egypt, Algeria, Morocco, Turkey, and Iran [22,23,24,25,26,27,28,29,30].

The UAE had very little data on the incidence or prevalence of fungal disease until 2024 [31]. Prior to this, there was only a single-center retrospective study on incidence rates of candidaemia that was published almost three decades ago [32]. Until 2024, no other dedicated epidemiological or clinical studies on fungal diseases had been published in the UAE.

The UAE national antimicrobial resistance (AMR) surveillance program [33] launched in 2015 and joined the WHO’s global AMR surveillance (GLASS) in 2018. The first-ever report of the UAE national AMR surveillance program was published in 2021, followed by the second report in 2022. Both reports included trends in antimicrobial resistance in *Candida* species only. No other fungal species were reported or surveyed [34,35].

The aim of this project is to determine an estimate of the burden of fungal diseases in the UAE, as a guide to local and national authorities on priority research and public health measures surrounding the increasing global threat of fungal infections.

## 2. Materials and Methods

### 2.1. Literature and Data Search

The grey and published medical literature was searched to obtain published data on the incidence and prevalence of each serious fungal disease in the UAE and surrounding countries in the gulf cooperation council (GCC). Where no data for the UAE or surrounding countries was found to inform estimates, international data was used.

### 2.2. Population and Underlying Data Sources

Country population data from 2020 was obtained from the federal statistics center to include both Emirati and foreign nationals [36]. Demographic data provided from the federal level was split by gender. Demographic data on age and a breakdown of Emirati versus foreign nationality was obtained from the Abu Dhabi Department of Health (DoH) 2017 statistics report [37]. DoH is the regulatory body of the health sector for the Emirate of Abu Dhabi, which includes the capital and largest city in the UAE, Abu Dhabi, and surrounding regions.

Data on persons living with HIV (PLHIV) were obtained from the UNAIDS organization report for 2022 [38]. There are no published data on documented cases of histoplasmosis, cryptococcal disease, *Pneumocystis* pneumonia (PCP), and esophageal candidiasis in the UAE. The disease burden of aforementioned HIV-related infections was estimated from the existing literature in the surrounding gulf countries Saudi Arabia [39], Qatar [40], Oman [41], and Bahrain [42].

Data on persons diagnosed with pulmonary tuberculosis (TB) annually for the last five years, the proportion of clinically diagnosed versus those diagnosed microbiologically, and annual deaths were obtained from the WHO TB estimated data [43] and the UAE’s national AMR surveillance data [44]. The aforementioned data were used to calculate the incidence and prevalence of CPA. CPA usually complicates other pulmonary conditions, especially pulmonary TB. To estimate the burden, we estimated CPA using TB data using the recently published model for India [45]. We then used a relative ratio of 1:5 of pulmonary TB with other underlying respiratory conditions associated with CPA, given the relative rarity of pulmonary TB in the UAE.

Women who have four or more episodes of vulvovaginal candidiasis (VVC) annually are defined as having recurrent vulvovaginal candidiasis. Using a 6% annual incidence rate in women >15–50 years, the prevalence of rVVC was calculated from multiple studies [46]. Fungal keratitis was estimated at 14/100,000, based on data from Egypt [47].

Data on the prevalence of asthma were obtained based on cross-sectional and survey-based studies conducted in the UAE, as a national registry for asthma is lacking [48,49]. The pooled prevalence data was used to estimate the prevalence of allergic bronchopulmonary aspergillosis (ABPA) and severe asthma and fungal sensitization (SAFS) complicating asthma. The prevalence of ABPA is assumed to be 2.5% of the prevalence of asthma in adults who present to secondary care based on a study from Saudi Arabia [50] and SAFS was estimated by assuming that 10% of asthmatic patients have severe asthma and sensitization to fungi occurs in approximately 33% of severe asthmatics.

Data on the prevalence of obstructive pulmonary disease (COPD) were also extrapolated from data based on cross-sectional and survey-based studies in the MENA region that included the UAE [51,52,53,54], as a national registry for COPD is lacking in the country. The prevalence of invasive aspergillosis is estimated to be 1.3% of hospitalized COPD patients [7,55].

The number of adult, pediatric, and neonatal intensive care beds and burn unit beds and the number of patients on chronic ambulatory peritoneal dialysis (CAPD) and hemodialysis were used to estimate the epidemiology of invasive candidiasis and *Candida* bloodstream infections. The incidence of candidaemia was determined (based on existing data) in patients with cancer, postsurgical procedures, critical care, and other immunocompromised patients at high risk. The annual incidence of *Candida* peritonitis was calculated using the assumption that if candidaemia occurs at a population rate of five cases per 100,000, and one-third are in ICUs, for every two patients with candidaemia in the ICU, there will be one patient with *Candida* peritonitis [56]. Given a higher overall rate of candidaemia, this may be an underestimate.

The prevalence of invasive aspergillosis was estimated by assuming that 10% of cases occurred in AML patients, and an equivalent number among all other hematologic malignancies, and 2.6% of cases from lung cancer [57,58] are combined with cases from COPD and cases after solid organ transplantation to generate a total. It is assumed that 1% of renal and 4% of liver transplant recipients develop invasive aspergillosis [59].

### 2.3. Frequency Estimates

Incidence and prevalence rates were estimated per 100,000 inhabitants in the UAE, including both the Emirati and non-Emirati population census.

### 2.4. Statistical Analysis

Descriptive statistics were used to summarize the data and expressed per 100,000 population. Sensitivity analysis was not performed, as the base estimates are themselves based on extrapolated data, so it would not be reliable.

## 3. Results

### 3.1. Country Demographics

The UAE is a multinational country with a population of 9.28 million in 2020 of both nationals and expatriates and a population growth rate of 1% annually [36]. There are more than 200 nationalities residing in the UAE, with the expatriate community outnumbering the population of UAE nationals. Males outnumber females owing to the large population of labor workers in the country. Indian nationals make up the largest expatriate community in the UAE (MOHAP). A federation of seven emirates, Abu Dhabi is the capital of the UAE and the largest city in the country with nationals making up 20% of its residence of whom 66.6% are under 30 years old in the 2018 statistics report. Expatriates are predominantly aged between 20 and 40 [37].

### 3.2. Health System Organization and National Antimicrobial resistance (AMR) Surveillance Program

The health sector in the UAE consists of three different healthcare systems that cover public and private healthcare facilities across all seven emirates. The health sector is well serviced, with density of primary healthcare facilities making up 3.7 per 10,000 population (2022) and a hospital bed density of 19.4 per 10,000 population (2020). Healthcare workers density in 2020 amounted to 28.8 per 10,000 for physicians, 63.3 per 10,000 for nurses, and 12.0 per 10,000 for pharmacists (UAE MOHAP-WHO health indicators).

The UAE national antimicrobial resistance (AMR) surveillance program consists of a network of 45 microbiology laboratories and 341 clinical surveillance sites across the country, spanning three different healthcare systems [33]. The first-ever report of the UAE national antimicrobial resistance (AMR) surveillance program was published in 2021, where AMR data on 482,312 patients were analyzed during the reporting period between 2010 and 2019, and only *Candida* species were included, representing 3336 isolates. The second UAE national AMR surveillance report was published in 2022, accounting for the surveillance period inclusive of 2020 [33,34,35].

### 3.3. Patient Populations

#### 3.3.1. HIV

The UAE is considered a low prevalence country according to the UNAIDS organization [38], with the UNAID report in 2022 reporting 1600 patients aged 15 and over living with HIV, <500 of them women and <100 deaths. Only 47% are on ART. These numbers only include UAE nationals and not expatriates living in the UAE.

#### 3.3.2. Tuberculosis

The WHO categorizes the UAE as a low endemic burden country for tuberculosis (TB), with an estimated TB incidence in 2022 of 0.76 per 100,000 population [43]. Numbers typically provided to the WHO by the UAE’s MOHAP include total TB cases amongst nationals, while data from the AMR surveillance database [44] include both nationals and non-nationals in the country for the year 2021 covering 95% of surveillance sites; the latter has been used for our estimations.

The AMR surveillance database reports a total of 881 TB cases (2021) and, estimating missing surveillance sites, we calculate the number to be 925. Of these 925, we estimate 10% are non-pulmonary TB, bringing up the total of pulmonary TB (PTB) to 832 cases in 2021. We assume most, if not all, are in non-HIV patients. The WHO 2022 data indicates that 87% of pulmonary cases are bacteriologically proven, bringing total cases of PTB to 957 cases. Including data from the AMR surveillance database and WHO estimates, we calculate a mortality rate of 5%.

#### 3.3.3. Malignancy and Transplantation

According to the UAE’s MOH 2019 statistics report, a total of 4633 newly diagnosed cancer cases (malignant and in situ) was reported to the UAE national cancer registry, of which 220 were categorized under leukemia, 215 cases under non-Hodgkin lymphoma, and 151 were categorized under lung cancer.

The UAE’s national program for organ donation and transplantation reported 2022 data to the international registry in organ donation and transplantation (IRODaT) [60], revealing a total of 149 kidney donations (deceased and living), 57 liver donations (deceased and living), and 6 deceased lung donations.

Bone marrow (hematopoietic stem cell) transplantation is in its infant stages in the UAE, with the first patient transplanted in 2020 [61]. Official reports by the UAE’s MOH are still being completed for the years 2020 and 2021 which will include updated numbers of recipients in the country. In one of eight licensed centers in the UAE, 18 pediatric patients have received a bone marrow transplant since March 2022, with numbers expected to reach 50 patients by the end of 2023 [62].

#### 3.3.4. Respiratory Populations (Asthma and COPD)

Based on local data extrapolated from a cross-sectional study and a survey-based study [47,48], asthma prevalence in adults in the country is estimated to range from 7.3 to 9.8%. We have assumed a 9.8% adult prevalence because this large study included both Emiratis and foreign nationals in an appropriate ratio, and reported prevalence in adults aged 20–44 years. Asthma control in the UAE is poor, with many not using maintenance inhaled corticosteroids [63]. The prevalence of COPD was reported to be as low as 1.9% in the largest MENA survey-based cross-sectional study [51], and as high as 3.7% [52] and 12.9% [53] in post-bronchodilator, spirometry-based studies in Abu Dhabi and Dubai, respectively. We have assumed a 3.7% population prevalence, and that 40% of these people are admitted to hospital with an exacerbation annually [54].

### 3.4. Invasive Candidiasis/Candidaemia

In parallel with the development and implementation of the UAE National AMR Surveillance program over the years, candidaemia data have been collected from a continuously increasing number of healthcare facilities in the UAE, beginning with 22 facilities in 2010 (6 hospitals, 16 centers/clinics, Abu Dhabi Emirate only), up to a total of 341 facilities in 2022 (90 hospitals, 251 centers/clinics, all seven Emirates). The 90 participating hospitals in 2022 represent 59.6% of all hospitals (n = 151) in the UAE, covering approximately 60% of the admitted patient population in the UAE. The 251 outpatient centers/clinics represent 9.2% of all relevant centers/clinics (n = 2730) in the UAE. In 2022, 654 cases of candidaemia were captured by the system, probably representing 1090 cases in the country (11.8 per 100,000). Of these, 7 (1.3%) and 28 (5%) of the 560 children with their age documented were from neonatal and pediatric patients, respectively. 

Table 1 shows the annual number of reported cases of candidaemia, as well as the estimated total (extrapolated) number of invasive candidaemia cases, based on a total UAE population of 9.28 million in 2020, and a coverage rate of 60% (hospitals).

The number of invasive candidiasis cases with a negative blood culture is likely to be 1.5 times higher (1635), given that blood culture is ~40% sensitive (i.e., negative blood culture results are common despite candidaemia being present) and antifungal prophylaxis is widespread in intensive care and complex patients in the UAE. As such, and based on the most recent (2022) data, the total annual incidence of invasive candidiasis and candidaemia in the UAE is estimated as 2725 (29.4/100,000).

Overall, 43.3% of patients with a positive blood culture for *Candida* spp. cultured were in ICUs. Among adults in ICUs, there were probably twice as many *Candida* bloodstream infection cases as intraabdominal *Candida* infection cases [7]. Data on ICU admissions between patients with *C. auris* versus other *Candida* spp. infections in the report revealed a statistically significant difference, where out of a total of 19,353 patients with non-*auris Candida* spp., 20.2% (3905 patients) were admitted to ICUs as opposed to 414/835 patients with *C. auris* (49.6%).

During the observation period (2010–2020) and for *Candida* spp. from all sources, a statistically highly significant shift, from *C. albicans* to non-*C. albicans* species, has been observed, with *C. albicans* accounting for 79% of all *Candida* spp. isolates in 2010, whereas in 2020 it was only 47%. The proportion of non-*Candida* albicans species among all *Candida* spp. accordingly increased from 21.4% (2010) to now 53.8% (2020) (Figure 2).

The observed increase over time of non-*albicans Candida* species is mostly due to an increase in the number of reported isolates of the following three non-*albicans Candida* species: *C. tropicalis*, *C. parapsilosis*, and *C. auris*. Whereas these three *Candida* species jointly accounted for only 14.9% (49/329) of all *Candida* spp. isolates reported in 2010, this percentage increased to 30.6% (1332/4355) in 2020 (Figure 3).

For blood cultures growing *Candida* species, a similar shift has been observed, with *C. albicans* accounting for 43.2% of all *Candida* spp. grown in 2010, decreasing to only 23.5% of all *Candida* spp. isolates in blood in 2022 (Table 2). During the same period, the proportion of non-*albicans* invasive *Candida* spp. increased from 56.8% (2010) to 81.2% (2022). The percentages do not always add up to 100% as patients may have more than one isolate in any given year (Table 2).

Differences in median of length of hospitalization between both groups was also statistically significant, with patients associated with *C. auris* infections accumulating a median length of stay of 33.5 days as opposed to 14 days for patients associated with non-*auris Candida* infections. Thomsen J et al. also calculated a mortality rate of 27.5% from a subset of patients with known admission and discharge dates in the *C. auris* group (47/171 patients) [31].

The trends for antifungal resistance in *Candida albicans* isolates were the only ones included in the 2019 report [34] revealing resistance of *C. albicans* to antifungal agents ranging from 1% for caspofungin and micafungin to 14% for voriconazole. Resistance of *C. albicans* to polyenes (amphotericin B) increased from 3% in 2010 to 17% in 2016 and then decreased to 4% in 2019. Triazole resistance has been increasing with resistance to fluconazole increasing from 1.7% (2010) to 4.6% (2019) and resistance to voriconazole increasing from 1.3% (2010) to 14.0% (2019). Resistance of *C. albicans* to echinocandins has shown decreasing trends, with resistance to caspofungin decreasing from 4.4% (2014) to 1.1% (2019) and resistance to micafungin decreasing from 3.6% (2014) to 1.4% (2019).

The 2022 national surveillance report [35] (inclusive of the 2020 reporting period) revealed that resistance of *C. albicans* to polyenes (amphotericin B) remained the same at 4.0% in 2020 compared to 2019 while resistance to fluconazole slightly decreased to 4.4% in 2020 compared to 4.6% in 2019 and resistance to voriconazole revealing a decrease to 6.0% (2020) from 14% in 2019. Resistance of *C. albicans* to echinocandins has maintained a decreasing trend in 2020, with resistance to caspofungin decreasing to 0.5% (2020) compared to 1.1% in 2019 and resistance to micafungin decreasing to 0.8% (2020) compared to 1.4% in 2019.

*Candida auris* was first reported in 2017 in the UAE [64], and included in the 2021 UAE national AMR report, with reported isolates increasing in frequency between 2010 and 2020 (n = 0 to n = 191). Extrapolating from the national AMR surveillance database, Thomsen J et al. [31] reported a total of 908 non-duplicate *C. auris* isolates from 2018–2021 (2018: n = 9; 2019: n = 93; 2020: n = 192; 2021: n = 614). *C. auris* isolates from urine sources constituted the leading sample size (280/908, 30.8%), followed by blood sources (248/908, 27.3%) skin and soft tissue sources (221/908, 24.3%), the respiratory tract (142/908, 15.6%), the genital tract (3/908, 0.3%), and cerebrospinal fluid (CSF) specimens (2/908, 0.2%).

### 3.5. Invasive Aspergillosis

Based on an estimated 6 patients with IA complicating lung cancer, 26 affecting transplant and haematological patients, and 477 patients developing IA in COPD (1.3% of COPD patients admitted to hospital with an exacerbation), our total estimated annual incidence is 505 patients or 5.4 per 100,000 (Table 3). These figures ignore other immunodeficiency states and patients in intensive care for reasons other than COPD or major immunosuppression, such as those with influenza. No data are collected nationally on species of *Aspergillus* causing infection, or antifungal susceptibility profiles. Mucormycosis and invasive aspergillosis can be mistaken, and only anecdotal case reports of mucormycosis are reported from the UAE [65,66], with fewer than 10 cases registered in the FungiScope^®^ registry (Table 3) [67].

### 3.6. Invasive Fungal Infections Associated with HIV

With only 1600 patients with HIV infection in the UAE, but <50% on ART, we estimate an at-risk population of 204 with respect to serious fungal infection. Cryptococcal meningitis is uncommon in the Middle East, estimated at 1.2% [68], so probably two cases annually. We assume another six cases in non-HIV patients, based on other country data [69]. *Pneumocystis* pneumonia (PCP) is more common, at 15.1% in Bahrain [42], so probably 15 HIV-related cases annually (Table 3). We have estimated incident PCP cases in non-HIV patients at 10 times as many, based on ratios from other high-income countries with low HIV prevalence. In the HIV population, there may be ~20 patients with esophageal candidiasis and more with oral candidiasis. Unless imported, we would anticipate no cases of histoplasmosis or talaromycosis.

### 3.7. Chronic Pulmonary and Allergic Aspergillosis and Fungal Asthma

Extrapolating from both WHO TB estimates and the national AMR surveillance TB database, we estimate the annual incidence of CPA in patients with pulmonary TB or thought to have pulmonary TB is around 298 (Table 3), with an estimated number of 10 deaths annually. We estimate the 5-year period prevalence (assuming a 1.5% annual attack rate of CPA after TB cure) is 668 patients in the context of presumed or documented pulmonary TB, with 57 deaths annually. Given the high number of COPD patients and other respiratory disorders, our total CPA prevalence estimate approaches 1002 cases.

Assuming that 9.8% of adults have asthma (nearly 755,000), and that 2.5% of these people have ABPA, we anticipate 14,635 (158/100,000) affected (Table 3). Severe asthma remains a problem in the UAE, with 23.8% of patients on biologicals in a recent survey of 458 Emiratis being managed in three hospitals in Dubai and Abu Dhabi [70]. The only data published on fungal skin test sensitization in Abu Dhabi were from 2012 in 62 adults with possible respiratory allergies [71]. Low frequency of sensitization to *Cladosporium* spp., *Aspergillus fumigatus*, *Ulocladium atrum*, *Alternaria alternata*, and *Penicillium* spp. were found. Assuming 33% have fungal sensitization in the worst 10% of adult asthmatics in terms of asthma control, an estimated 193,318 have SAFS (208/100,000) (Table 3).

### 3.8. Superficial Fungal Infection

#### 3.8.1. Vulvovaginal Candidiasis (VVC)

There is only one single-center retrospective study on VVC in the UAE demonstrating an incidence of VVC significantly increasing from 10.8% in 2005 to 17.6% in 2011 with an average prevalence of 13.9% [72]. We have conservatively assumed that 6% of the 4 million adult women (aged 15–50) have recurrent VVC, a total of 190,330 (Table 3). Hamad et al. found the percentages of species implicated to be *C. albicans* was 83%, *C. glabrata* was 16.5% and *C. tropicalis* was 1.2% [72].

#### 3.8.2. Tinea Capitis

There is only one single-center retrospective study on tinea capitis in the UAE conducted between 1981 and 1988, reporting 234 dermatophyte isolates resulting in tinea capitis [73]. We have not been able to estimate its prevalence, but it is likely to be very low.

#### 3.8.3. Fungal Keratitis

No studies have been conducted on incidence or prevalence of fungal keratitis in the UAE. Fungal keratitis annual incidence was estimated at 1336 or 14/100,000 based on data from Egypt [47] (Table 3).

### 3.9. Neglected Fungal Tropical Diseases and Entomophthoromycosis

We found no reports of eumycetoma, chromoblastomycosis, or sporotrichosis from the UAE—we suspect these are rare infections, and probably not endemic. Basidobolomycosis is observed in the Middle East, predominantly in children, but there are no reported cases we could identify. One case of entomophthoromycosis is reported from the UAE [74]. Cerebral abscess caused by *Rhinocladiella mackenziei* has been reported once [75] and is occasionally seen in the Middle East.

## 4. Discussion

This is the first attempt to estimate the incidence and prevalence of fungal diseases in the UAE, where in the absence of epidemiological or clinical studies, we have derived some data from the UAE national antimicrobial resistance (AMR) surveillance program and supplemented data with estimates provided by the UAE’s MOHAP and the WHO. The most substantial fungal disease issue is recurrent VVC, followed by fungal asthma, the collective term for ABPA and SAFS.

The UAE national AMR surveillance program [33] is representative for public sector health facilities in the UAE, as it covers all public health institutions, and is highly representative for private health facilities, except for the emirates of Ajman, Um Al Quwain, and Fujairah. The AMR surveillance data are also highly representative for inpatients and intensive care unit (ICU) patients, with 58% of institutions housing both patient populations included in the surveillance. A total of 2730 ambulatory healthcare clinics and centers are enrolled in the national AMR surveillance which contributes to good representation of the outpatient population as well.

Despite the AMR surveillance program’s good representation of the patient populations in the UAE, only *Candida* species among the fungi are represented in the surveillance database. However, this representation of *Candida* species in the surveillance database did reveal important insights on resistance patterns and a significant shift from *C. albicans* to non-*albicans Candida* spp. [34,35]. The recent study published by Thomsen et al. [31] revealed alarming trends in the resistance patterns of *C. auris* in the UAE, analyzing 908 *C. auris* isolates from the 2018–2021 observation period and offered some demographic data associated with *C. auris* infection.

Apart from a 1995–2001 single-center retrospective study [32] looking at associated demographic and clinical data in candidaemia, there have been no studies exploring patient risk factors with invasive candidiasis or other fungal infections in the UAE. That study evaluated 60 episodes of candidaemia through case retrieval and found that 65% of patients had a diagnosis of malignancy, and *C. albicans* accounted for 45% of the isolates. Crude mortality in the sample population was 50%, with 30% of mortality attributable to candidaemia.

Recurrent VVC is a much more common problem than generally realized, with a major impact on women’s self-esteem and their relationships. The point prevalence of VVC of 10.8% in 2005 rose to 17.6% in 2011, speaking to a substantial numerical problem [42]. Little data are available on the proportion of these patients infected with *C. albicans* or the more challenging fluconazole-resistant *C. glabrata*.

The UAE is considered a low HIV prevalence country [58], and with only 1600 patients with HIV infection in the UAE, but <50% on ART, we estimate an at-risk population of 204 with respect to serious fungal infection, with cryptococcal meningitis estimated at 2 cases annually, 15 cases of PCP annually, and 20 cases of esophageal candidiasis in the HIV population.

Likewise, the UAE is also considered a low endemic country for TB, with an estimated WHO TB incidence in 2022 of 0.76 per 100,000 population [43]. Using both WHO estimates and the national AMR surveillance TB data, we calculated a total CPA prevalence estimate of 1002 cases. CPA prevalence based on other Middle Eastern countries (notably Kuwait) is 21.3 per 100,000 prevalence [30] with 1975 cases annually.

Deriving our estimates from local data on asthma and COPD in the UAE, we estimate 14,770 cases of ABPA (157/100,000) in the UAE.

Based on local data on malignancy, transplant, and COPD, we estimate a total annual incidence of invasive aspergillosis of 509 patients or 5.4 per 100,000. These figures are not inclusive of other immunodeficiency states and patients in ICUs for reasons other than COPD or with major immunosuppression, so do not include cases linked to influenza or COVID-19.

This study attempts to estimate fungal infections in the UAE despite a lack of clinical and epidemiological data. There is currently no fungal reference laboratory in the UAE. The paucity of data on fungal isolates in the UAE reflects a grave deficiency in fungal diagnostics for identification and species susceptibility testing, a lack of local expertise in clinical mycology, and little to no research output in the field locally and regionally.

## 5. Conclusions and Future Perspectives

This is the first study on the estimated burden of fungal infections in the United Arab Emirates. This report might be an underestimate of the magnitude of the problem due to lack of chronic disease registries in the country. Our paper highlights the urgent need for more surveillance of invasive fungal pathogens and diseases and a dedicated fungal reference laboratory in the country to improve diagnostics and aid in national surveillance efforts.

## Figures and Tables

**Figure 1 jof-10-00353-f001:**
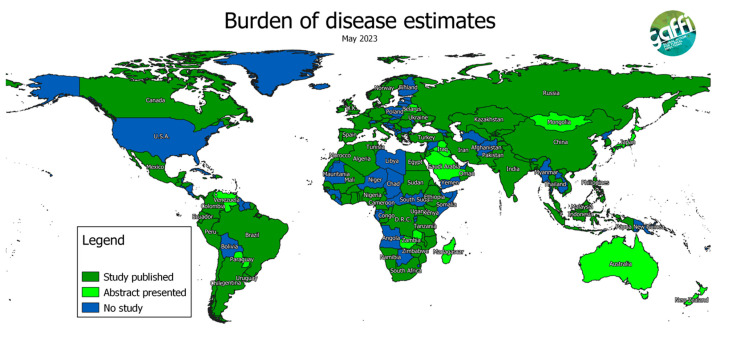
A Map showing completed country estimates of fungal diseases by May 2023 [20].

**Figure 2 jof-10-00353-f002:**
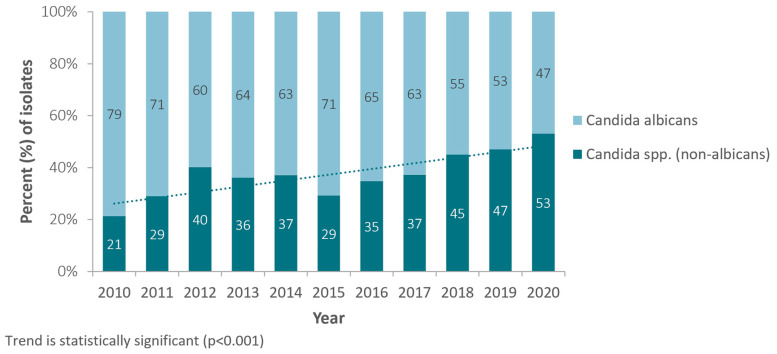
Annual trend for percentage of *Candida* (non-*albicans*) isolates, among all *Candida* isolates (*Candida* spp.), United Arab Emirates, 2010–2020.

**Figure 3 jof-10-00353-f003:**
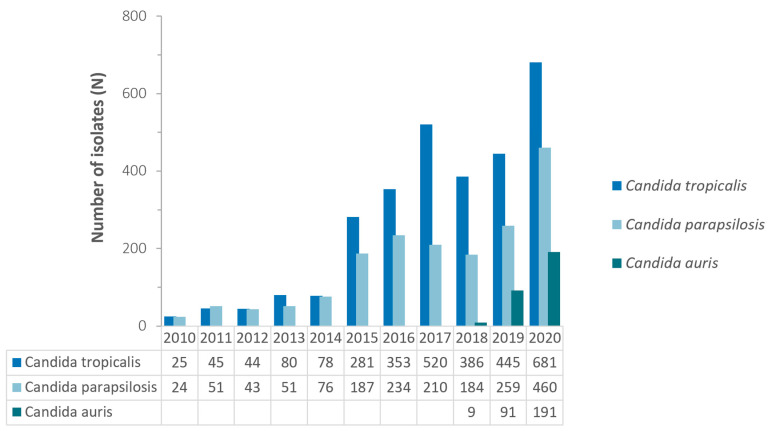
Annual trend for number of selected non-*albicans Candida* spp., United Arab Emirates, 2010–2020.

**Table 1 jof-10-00353-t001:** National surveillance of candidaemia, UAE, 2010 to 2022 (all age groups, all patient locations), reported cases versus estimated/extrapolated cases.

Number of Patients with Candidaemia	% of UAE Population Covered	2010	2011	2012	2013	2014	2015	2016	2017	2018	2019	2020	2021	2022
3517 (as reported)	60	44	73	76	85	117	188	213	232	260	278	482	815	654
5861 (estimated, by extrapolation)	100	73	122	127	142	195	313	355	387	433	463	803	1358	1090

**Table 2 jof-10-00353-t002:** National surveillance of invasive candidaemia, UAE, 2010–2022, by species (*Candida albicans* versus *Candida* spp. (non-*albicans*)).

Number of Patients with Invasive Candidaemia	%	2010	2011	2012	2013	2014	2015	2016	2017	2018	2019	2020	2021	2022
3517 (as reported, total)	100	44	73	76	85	117	188	213	232	260	278	482	815	654
908 (*Candida albicans*)	24	19 (43.2)	27 (37.0)	32 (42.1)	23 (27.1)	36 (30.8)	46 (24.5)	56 (26.3)	62 (26.7)	76 (29.2)	75 (27.0)	120 (24.9)	182 (22.3)	154 (23.5)
2706 (*Candida* spp. (non-*albicans*))	76	25 (56.8)	48 (65.8)	45 (59.2)	62 (72.9)	82 (70.1)	144 (76.6)	165 (77.5)	174 (75.0)	190 (73.1)	206 (74.1)	379 (78.6)	655 (80.4)	531 (81.2)

**Table 3 jof-10-00353-t003:** Estimated incidence and prevalence of fungal disease in the UAE.

Infection	Number of Infections Per Underlying Disorders Per Year						Rate/100 K	Total Burden
	Incidence or Prevalence	None	HIV/ AIDS	Respiratory	Cancer/Transplant	ICU		
Candidaemia	I	-	-	-	554	536	11.8	1090
Candida peritonitis	I	-	-	-	-	268	2.9	268
Invasive aspergillosis	I	-	-	6	26	473	5.4	505
Mucormycosis	I	-	-	-	19	-	0.2	19
Chronic pulmonary aspergillosis	P	-	-	1002	-	-	10.8	1002
ABPA	P	-	-	14,635	-	-	158	14,635
SAFS	P	-	-	19,318	-	-	208	19,318
Cryptococcal meningitis	I	2	2	-	4	-	0.1	8
*Pneumocystis* pneumonia	I	-	15	-	150	-	1.8	165
Esophageal candidiasis	I		20				0.2	20
Recurrent vaginal candidiasis (>4 times/year)	P	190,329	-	-	-	-	4102 *	190,329
Fungal keratitis	I	1336	-	-	-	-	14	1336
**Total burden estimated**		**191,665**	**37**	**34,955**	**753**	**1277**		**228,695**

* Only female population.

## Data Availability

There are no additional data sources available, as most of the data is taken from the published literature. Access to the National Antimicrobial Resistance database is possible via application to the Ministry of Health and Prevention.

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
