# Peer review of "Mapping the Burden of Fungal Diseases in the United Arab Emirates"

_jof, 2024, doi:10.3390/jof10050353_

Round 1

Reviewer 1 Report

I consider that the introduction and the results should be restructured. In the attached file I detail the things that I consider should be corrected. 

Regarding the discussion of this work, I consider that it is not possible to talk about incidences and indicate that Candida infections are the most common, when there is no complete database, if so, they should focus the study only on Candida spp.

Table 2 and 3 are not cited in the text. Neither is figure 1. 

In the attached file I detail each of the comments by sections. 

Author Response

Responses to highlighted comments:

UAE, TB and COPD spelt out

Reference 1 focusses on species extinction – mostly amphibians and plants but some other vertebrates. It made the cover of Nature, as it was the first paper to evaluation compare bacterial, virus and fungal causes of species extinctions.

Para 1 text writing improved

Figures and tables listed in the text.

Section 3.1 Country demographics. We disagree that this information is irrelevant as it population structure pertains to some fungal disease risk categories and especially imported fungal diseases and people sent back to their country of origin because they are ill.

Section 3.2 Likewise we disagree about the structure of the health service. This sets the scene for diagnostic capacity and some risk groups such as advanced cancer therapies, transplantation and ICU care.

Section 3.4 We prefer the British spelling of candidaemia

Table 2 C. albicans italicised

Conclusions now called  ‘Conclusions and future perspectives’

Reviewer 2 Report

The prevalence of fungal infections worldwide is on the rise, however, data regarding the etiology of such infections are largely missing. The burden of fungal disease in UAE is interesting and important new knowledge. In the manuscript etiological agents, if known, of infections such as recurrence vaginal candidiasis, and keratitis should be added. If not it should be stated in the text and the reason should be explained, eg. lack of specific data, no identification performed, and so on. Also to etiology of candidaemia should be specified.

No specific corrections, except for the listed major concerns were detected.

Author Response

The prevalence of fungal infections worldwide is on the rise, however, data regarding the etiology of such infections are largely missing. The burden of fungal disease in UAE is interesting and important new knowledge. In the manuscript etiological agents, if known, of infections such as recurrence vaginal candidiasis, and keratitis should be added. If not it should be stated in the text and the reason should be explained, eg. lack of specific data, no identification performed, and so on. Also to etiology of candidaemia should be specified.

Thank you. We have added data on causative fungi as much as we can based on the data we have, including VVC. We plan a much fuller paper on the causative agents of deep Candida infections.

Reviewer 3 Report

The study conducted by Al Dhaheri and colleagues presents estimates of the fungal disease burden in the United Arab Emirates. Authors managed to explore all relevant fields in fungal diseases and presented estimates along with real life data obtained from local and national registries. From my standpoint it is an interesting study, with a solid methodological background that gives a relevant contribution to current epidemiological investigation regarding fungal diseases. References are up to date, tables and figures are clear and easy to interpret. Another strong point of this study is the evaluation and description of antifungal resistance rates and trends that adds more value to this already knowledgeable study.

In the end this is a well-conducted study presenting useful information. I would suggest authors to further evaluate and discuss the following minor issues:

Line 284-285. Authors mention only the most frequently isolated species, but it would be extremely important to present also related percentages across the entire study period, if the data are available, both in the text and in table 2.

Line 302-310. Antifungal resistance in Candida spp has only been presented for C. albicans, however it is well known that the burden of antifungal resistance associated with Candida spp nowadays is carried out mostly by non-albicans species, apart from C. auris. I would suggest presenting data regarding antifungal resistance for the other species reported, at least for the two most commonly isolated from candidemia.

Line 454-458. The following sections must be filled:  Institutional Review Board Statement, Informed Consent Statement, Data Availability Statement.

Author Response

Major comments

The study conducted by Al Dhaheri and colleagues presents estimates of the fungal disease burden in the United Arab Emirates. Authors managed to explore all relevant fields in fungal diseases and presented estimates along with real life data obtained from local and national registries. From my standpoint it is an interesting study, with a solid methodological background that gives a relevant contribution to current epidemiological investigation regarding fungal diseases. References are up to date, tables and figures are clear and easy to interpret. Another strong point of this study is the evaluation and description of antifungal resistance rates and trends that adds more value to this already knowledgeable study.

Detail comments

In the end this is a well-conducted study presenting useful information. I would suggest authors to further evaluate and discuss the following minor issues:

Line 284-285. Authors mention only the most frequently isolated species, but it would be extremely important to present also related percentages across the entire study period, if the data are available, both in the text and in table 2.

Response: We have added this detail in a new figure 2 showing the data from 2010 to 2020.

Line 302-310. Antifungal resistance in Candida spp has only been presented for C. albicans, however it is well known that the burden of antifungal resistance associated with Candida spp nowadays is carried out mostly by non-albicans species, apart from C. auris. I would suggest presenting data regarding antifungal resistance for the other species reported, at least for the two most commonly isolated from candidemia. 

Response: We have added this detail in a new figure 3 showing the data from 2010 to 2020.

Line 454-458. The following sections must be filled:  Institutional Review Board Statement, Informed Consent Statement, Data Availability Statement.

Response: All added.

Reviewer 4 Report

The manuscript is important in order to know the prevalence of fungal infections in the United Arab Emirates. In this sense, to provide an attractive article the results should be improved to understand better all the factors which fungal infection are involve. The contribution of this study will be in the benetif of the UAE and to global continents.

In several parts and sections of the manuscript, the authors use abbreviations. However, several of them have not been introduced. The authors should write first and then use it in abbreviation. Please revise carefully.

Author Response

The manuscript is important in order to know the prevalence of fungal infections in the United Arab Emirates. In this sense, to provide an attractive article the results should be improved to understand better all the factors which fungal infection are involve. The contribution of this study will be in the benefit of the UAE and to global continents.

In several parts and sections of the manuscript, the authors use abbreviations. However, several of them have not been introduced. The authors should write first and then use it in abbreviation. Please revise carefully.

Response: Thank you – we have attempted to remove all’ loose’ abbreviations.

Round 2

Reviewer 4 Report

I agree with the new version provided by the authors

No comments